# Acidic Gas Determination Using Indium Tin Oxide-Based Gas Sensors

**DOI:** 10.3390/s24041286

**Published:** 2024-02-17

**Authors:** Kaiyan Peng, Qiang Li, Mingwei Ma, Na Li, Haoran Sheng, Haoyu Li, Yujie Huang, Feng Yun

**Affiliations:** 1Key Laboratory of Physical Electronics and Devices for Ministry of Education and Shaanxi Provincial Key Laboratory of Photonics & Information Technology, Xi’an Jiaotong University, Xi’an 710049, China; pkymail@stu.xjtu.edu.cn; 2School of Electronic Science and Engineering, Xi’an Jiaotong University, Xi’an 710049, China; mamingwei934@163.com (M.M.); baekhyun_shr_2001@stu.xjtu.edu.cn (H.S.); msn010@stu.xjtu.edu.cn (H.L.); hyj2216224437@stu.xjtu.edu.cn (Y.H.); fyun2010@mail.xjtu.edu.cn (F.Y.); 3Northwest Survey & Planning Institute of National Forestry and Grassland Administration, Xi’an 710048, China; superman1232004@126.com

**Keywords:** indium tin oxide, SO_2_ detection, NO_2_ detection, mechanism of gas-sensitive detection

## Abstract

This work has presented gas sensors based on indium tin oxide (ITO) for the detection of SO_2_ and NO_2_. The ITO gas-sensing material was deposited by radio frequency (RF) magnetron sputtering. The properties of gas sensing could be improved by increasing the ratio of SnO_2_. The response characteristics of the gas sensor for detecting different concentrations of NO_2_ and SO_2_ were investigated. In the detection of NO_2_, the sensitivity was significantly improved by increasing the SnO_2_ ratio in ITO by 5%, and the response and recovery time were reduced significantly. However, the sensitivity of the sensor decreased with increasing SO_2_ concentration. From X-ray photoelectron spectroscopy (XPS) analysis, the gas-sensitive response mechanisms were different in the atmosphere of NO_2_ and SO_2_. The NO_2_ was adsorbed by ITO via physisorption but the SO_2_ had a chemical reaction with the ITO surface. The gas selectivity, temperature dependence, and environmental humidity of ITO-based gas sensors were systematically analyzed. The high detection sensitivity for acidic gas of the prepared sensor presented great potential for acid rain monitoring.

## 1. Introduction

Acid rain is one of the three major global environmental hazards, with a pH far below that of normal rainfall, which is largely detrimental to ecosystems, human health, and buildings [1]. Since acid rain is mainly generated by the reaction of NO_2_ and SO_2_ with water vapor, gas sensors are widely used to monitor the concentration of NO_2_ and SO_2_ in the atmosphere and timely control the emission of acidic substances. In recent years, metal oxide semiconductor gas sensors have been widely used for acidic gas detection due to their low cost, high sensitivity, and short response and recovery time [2,3,4,5]. Indium oxide (In_2_O_3_) and tin oxide (SnO_2_) are the most common sensitive materials used for acid rain gas detection. Excellent response to 0.1 ppm NO_2_ has been achieved by utilizing bimetallic Pd/Pt modification to prepare SnO_2_ nanowires. However, the device needs to operate at temperatures as high as 300 °C [6]. An ITO-ZnO composite film was prepared to detect NO_2_, achieving a lower detection limit of 70 ppm at an operating temperature of 170 °C but with a response time as long as 70 s [7]. A SnO_2_ thin film was prepared to detect SO_2_ at an operating temperature of 170 °C; however, the lower detection limit was 500 ppm with a low sensitivity of 1.3 [8]. It has been shown that the selectivity and sensitivity of the gas sensor can be further improved by changing the doping elements, doping concentration, and surface morphology of the sensitive material. At the same time, the effective life of the device for corrosive gas detection will be reduced at high operating temperatures, so it is necessary to further reduce the operating temperature. 

Indium tin oxide (ITO), as a typical wide bandgap metal oxide semiconductor, is suitable for gas sensing due to its low resistivity and strong mechanical hardness. ITO-based sensors have been used to detect various gases, such as H_2_, CO, NH_3_, methanol, ethanol, benzene gases, etc. [9,10,11,12,13,14,15,16,17]. We have fabricated ITO materials with different morphologies (film, nanoparticles, nanorods, and nanowires) by sputtering [18]; the RS of the sensor with appropriate ITO nanowires was 235.6 at the ethanol gas concentration of 400 ppm, and the response and recovery times are 10 s and 12 s [19]. Meanwhile, the excellent corrosion resistance gives tin oxide and its compounds a significant advantage in the detection of corrosive gas [20].

In this study, ITO gas-sensitive materials with different In_2_O_3_/SnO_2_ ratios were prepared, characterized, and used to detect NO_2_ and SO_2_ with various concentrations. By increasing the SnO_2_ ratio in ITO film, the gas-sensitive response of ITO for the detection of SO_2_ and NO_2_ has been significantly improved. From the results, ITO-based gas sensors showed a decrease in sensitivity with increasing concentration in detecting SO_2_, which was different from the usual law of gas-sensitive detection, and the mechanism behind this phenomenon was studied. At last, the gas selectivity and temperature dependence of ITO-based sensors were also researched.

## 2. Methods

In this work, ITO films were prepared on the surface of the ceramic tubes using a radio frequency (RF) magnetron sputtering system, and then the ceramic tubes were soldered to the electric circuit to complete the fabrication of the gas sensors. The ceramic tubes were made of aluminum oxide with gold as the electrode material, and they were threaded with copper wire for heating. In the RF sputtering process, ITO target materials with In_2_O_3_/SnO_2_ ratios of 90:10 (wt%) and 85:15 (wt%) were used, respectively, under the argon gas flow rate of 20 sccm, the RF power of 45 W, the sputtering temperature of 27 °C, and the sputtering time of 98 min.

The sensitivity was defined as *R_g_*/*R_a_* and *R_a_*/*R_g_* for the detection of the oxidizing gas NO_2_ and the reducing gas SO_2_, respectively. Response time was defined as the time required for the sensor to complete 90% of its response after exposure to a specific concentration of target gas, and recovery time was defined as the time required for the sensor to recover 90% of its response after returning to the air. Maintain a constant humidity of 35%RH in the testing environment for all gas sensors.

Gas adsorption is a constantly changing process, so the test can be divided into static gas reaction process and dynamic gas reaction process. In the static process, a specific concentration of gas was filled in the chamber and then turned on the gas-sensitive test system. Static gas-sensitivity testing is used to monitor the macroscopic changes in sensor resistance caused by carriers’ concentration. The resistance of the gas sensor was changed by the concentration of carriers through the adsorption, desorption, and ionization reaction with gas. In the dynamic process, the gas-sensitive test system was turned on first, and then gas filled the chamber until a specific concentration. Dynamic testing is used to monitor the change in resistance of gas-sensitive materials when they make contact with a gas.

For the characterization of gas-sensing materials, scanning electron microscopy (SEM) was conducted on an FEI Quanta 250FEG (Hillsboro, OR, USA), and X-ray diffraction (XRD) was conducted on a Bruker D8 ADVANC (Karlsruhe, Germany). X-ray photoelectron spectroscopy was conducted on a Thermo Scientific Escalab 250Xi photoelectron spectrometer (Waltham, MA, USA), and we calibrated the result with the binding energy of elemental carbon, which is 284.6 eV, when we analyzed the test results. Atomic force microscopy (AFM) was conducted on a Shimadzu SPM-9700HT (Kyoto, Japan).

## 3. Results and Discussion

### 3.1. Effect of Sn Doping on the Morphology of ITO Films

The SEM results of ITO films prepared by magnetron sputtering were shown in Figure 1a,b; the size of the particles on the surface of the ITO film with In_2_O_3_/SnO_2_ ratio of 90:10 was significantly smaller than that of 85:15, and the number of particles per unit area also increased significantly with the increase of SnO_2_ concentration. The AFM characterization results of ITO films are shown in Figure 1c,d; with the increase in SnO_2_ concentration from 10% to 15%, the surface roughness (root-mean-square error) of the ITO films decreased from 2.34 nm to 1.95 nm. The results from SEM and AFM reveal that the ITO film with 15% SnO_2_ concentration exhibited a smaller surface particle size compared to the ITO film with 10% SnO_2_ concentration, which is attributed to the larger specific surface area. On one hand, increasing the specific surface area could provide more contact sites for the interaction between ITO and gas molecules, leading to the improvement of gas-sensitive performance [21]. On the other hand, for ITO films with smaller surface particles, the gas-sensitive performance improved because the portion of the depletion region increased [22].

### 3.2. Effect of SnO_2_ Ratio on Gas-Sensitive Properties

Figure 2a shows the gas-sensitive properties of ITO with different SnO_2_ concentrations under NO_2_ gas conditions, and the gas-sensitive properties were significantly improved with the increase in SnO_2_ concentration (the mass percent of SnO_2_ from 10% to 15%). The working temperature of the gas sensor was 240 °C. In Figure 2b, the sensitivity increased from 1.7 to 2.5 under 5 ppm NO_2_ and increased from 2 to 2.8 at 10 ppm NO_2_. In Figure 2d, when the NO_2_ was 5 ppm, the response time decreased from 20 s to 8 s, and the recovery time decreased from 110 s to 74 s. For ITO with a SnO_2_ concentration of 15% in Figure 2c, as the SO_2_ concentration rose from 5 ppm to 10 ppm, the response time decreased from 8 s to 6 s, and the recovery time decreased from 74 s to 61 s.

Figure 3a shows the gas-sensitive properties of ITO with different SnO_2_ concentrations under SO_2_ gas conditions, and the gas-sensitive properties of ITO were improved with the increase in SnO_2_ concentration. The same working temperature of 240 °C was used. Compared to under NO_2_ conditions, the amplitude of change was relatively small. In Figure 3b, the sensitivity increased from 1.5 to 1.9 under 5 ppm SO_2_. For 10 ppm, the sensitivity increased from 1.4 to 1.8. From Figure 3d, at about 5 ppm SO_2_, the response time decreased from 96 s to 66 s, and the recovery time decreased from 77 s to 58 s. When the ITO with SnO_2_ concentration increased to 15%, the response time and recovery time were 66 s and 58 s under 5 ppm SO_2_, respectively. As the SO_2_ gas changed from 5 ppm to 10 ppm, the response time and recovery time decreased to 42 s and 52 s.

The resistance of the ITO-based gas sensor decreased under oxidizing gas SO_2_ and increased under reducing gas NO_2_. For the detection of both SO_2_ and NO_2_, the gas-sensitive properties improve by improving the SnO_2_ concentration. ITO showed higher sensitivity and shorter response and recovery time in 5 ppm NO_2_ compared with 10 ppm NO_2_. However, for the detection of SO_2_, the sensitivity decreased as the concentration of SO_2_ increased from 5 ppm to 10 ppm, which was different from the usual law of gas-sensitive detection, and further experiments were conducted to verify this phenomenon by expanding the range of measured concentrations.

### 3.3. Gas-Sensitive Response Characteristics for NO_2_ and SO_2_

The response curves of the ITO gas sensor in 5–100 ppm NO_2_ gas were shown in Figure 4a; the sensor exhibited excellent gas-sensitive characteristics. The ITO-based sensor has high sensitivity in low concentrations of NO_2_; with increasing NO_2_ concentration, the sensitivity and response time increased, and the recovery time reduced (Figure 4c,e). The response curves of the sensor in SO_2_ with the concentration range of 5–100 ppm were shown in Figure 4b. Unlike the detection of NO_2_, the ITO-based gas sensor exhibited a decrease in sensitivity with increasing SO_2_ concentration when the concentration increases from 5 ppm to 100 ppm (Figure 4d,f). All the gas sensors were operated at the same working temperature of 160 °C.

The relevant gas-sensitive materials and results of testing NO_2_ and SO_2_ in recent years have been summarized in Table 1 and Table 2. The use of ITO materials to measure 5 ppm NO_2_ and 5 ppm SO_2_ achieved good experimental results, especially in terms of response time and recovery time, and are the best ones currently, fully demonstrating the potential of ITO materials as gas-sensitive materials.

In order to verify that the phenomenon occurring in the detection of SO_2_ is not a random phenomenon due to errors or other reasons, we changed the test method for SO_2_ gas at 10 ppm and 5 ppm. Unlike the dynamic test method used in the previous experiments, the static test was used to calculate and analyze the sensitivity of the sensor for low-concentration detection of SO_2_, and the results are shown in Figure 5. The values of ΔV_1_ and ΔV_2_ for the ITO gas sensor in a 5 ppm SO_2_ environment were 0.21 V and 0.32 V, respectively (Figure 5a), and the values of ΔV_3_ and ΔV_4_ for the ITO gas sensor in a 10 ppm SO_2_ environment were 0.043 V and 0.036 V, respectively (Figure 5b). The voltage change in the sensor in a 10 ppm SO_2_ environment was almost an order of magnitude smaller than the voltage change at 5 ppm. These results indicate that the ITO-based gas sensors had stronger gas-sensitive response in lower concentrations of SO_2_ because of the stronger electron transfer ability.

### 3.4. Effect of Sn Doping on Oxygen Vacancy Concentration

As an n-type semiconductor, the gas-sensing performance of ITO can be improved by increasing the concentration of oxygen vacancies [37]. Oxygen vacancies are common defects in ITO, and the defects are ideal adsorption sites for target molecules on the surface of gas-sensitive materials. When the sensor is in contact with air, the adsorbed O_2_ forms a peroxide structure, and once the O_2_ molecules are adsorbed on the top of the oxygen vacancy sites, the O atoms in the O_2_ molecules can spontaneously fill the oxygen vacancy defects through the unobstructed dissociation process. A depletion layer will be formed on the surface of the n-type semiconductor, and the concentration of the carriers will decrease, and thus the conductance of the sensor will decrease [38]. The adsorption energy of the measured gas at the oxygen vacancy site will be significantly increased. The adsorbed gas molecules may be dissociated into an O atom and other molecules after adsorption on the surface of the sensitive material. The dissociated O atoms from the measured gas will fill the oxygen vacancy sites, while the remaining gas molecules can be easily desorbed on the defect-free surface by an exothermic process, thus completing the electron transfer process [39].

When ITO is used as a gas-sensitive material, the content of oxygen inside the material will directly affect the gas-sensitive performance of the device. On one hand, since oxygen will first be absorbed on the surface of the sensitive material, it will be ionized into different states of adsorbed oxygen ions. When the sensor works, oxygen ions react with the target gas to produce carriers. Therefore, the adsorbed oxygen content will determine the concentration of adsorbed oxygen ions generated on the surface of the material. The higher the concentration of adsorbed oxygen ions, the more electrons will be exchanged when the material undergoes a gas-sensitive reaction, and the higher the sensitivity will be. On the other hand, the oxygen content in the material is partly in the form of lattice oxygen and partly in the form of oxygen vacancies caused by defects; thus, the higher the oxygen vacancy concentration, the more adsorbable sites for gas molecules and the higher the sensitivity.

Oxygen vacancies are defects brought about by Sn^4+^ doping in the indium oxide lattice, and the Sn^4+^ doping concentration is proportional to the oxygen vacancy concentration. The oxygen content in ITO films with different indium/tin ratios was fitted by peak splitting in the XPS analysis, and the fitting results are shown in Figure 6a,b. Two curves existed within the fitted peaks, corresponding to binding energies of 529.4 eV and 531.1 eV, respectively. The peaks at 529.4 eV were assigned to adsorbed oxygen, and the peaks at 531.1 eV could be assigned to lattice oxygen [19]. The results of integrating the different peak areas were calculated in Table 3. The content of adsorbed oxygen and lattice oxygen became larger with SnO_2_ ratio increases, which was the reason for the ITO-based gas sensor with 15% SnO_2_ having a better gas-sensing performance.

### 3.5. Analysis of Gas-Sensing Mechanism

In general, adsorption can be divided into physical and chemical adsorption [40]. The sensitivity of the prepared ITO-based gas sensor increases with the increase in concentration when detecting NO_2_, and the gas-sensitive reaction mechanism is as follows. When the ITO film was exposed to NO_2_, the gas molecules adsorbed to the surface of the material rob electrons from the material and cause the energy bands to bend near the contact surface. The thickness of the depletion layer on the surface of the material increased, and then the potential barrier at the grain boundary rose. Last, the resistance of the ITO gas-sensitive material increased. The NO_2_ molecules did not react chemically with the ITO material; only free electron transfer occurred. So, the adsorption between NO_2_ and the ITO material could be considered as physical adsorption.

Firstly, the NO_2_ gas molecules were adsorbed on the surface of the ITO material; then, the molecules captured free electrons from the conduction band of the ITO film, and NO_2_ was formed. The gas-sensitive reaction process can be described as Equations (1) and (2).
(1)$$O_{2}+2e^{-}\to 2O^{-}$$
(2)$$NO_{2}+e^{-}\to N{O_{2}}^{-}$$

Unlike the detection of NO_2_, the sensitivity of the prepared ITO gas sensors decreased with increasing SO_2_ concentration from the static and dynamic test results. The XPS analysis on the surface of an ITO gas sensor that had been used to detect SO_2_ was employed to check the reaction process (Figure 6c,d). The integration of the S element content revealed that the S element content was only 2.51%, and XPS results are shown in Figure 6c,d. The S2p spectrum exhibited two contributions, 2p3/2 and 2p1/2, located at 161.3 eV and 162.2 eV, respectively, which could be assigned to In_2_S_3_ [41]. The 3d5/2 and 3d3/2 peaks were observed at 168.9 eV and 169.2 eV, respectively (Figure 6d), which could be assigned to SnSO_4_ [42]. XPS results revealed that the response mechanism of the ITO-based gas sensor for the detection of SO_2_ involved chemical reactions, which led to the anomalous gas-sensing response.

Since the Sn-O bond in the ITO structure is more stable than the In-O bond, the SO_2_ gas molecules first reacted with In_2_O_3_ to form SO_3_. As the concentration of SO_2_ increases, the In_2_O_3_ that could take part in the reaction was completely reduced to In_2_S_3_, and SnO_2_ was further involved in the reaction with SO_2_. This was the reason for the different S-containing compounds on the surface of the samples at different concentrations exhibited in the XPS results. Meanwhile, the generated SO_3_ further reacted with adsorbed oxygen anions on the sensor surface, and the electrons released during the reaction returned to the conduction band of the ITO material. In a relatively low concentration of SO_2_, more In_2_S_3_ was formed as the SO_2_ concentration increased, reducing the electrons in the conduction band of the ITO material. Further increasing the concentration of SO_2_, the formed SO_3_ combined with adsorbed oxygen ions and reacted with SnO_2_ to form SnSO_4_, which reduced the concentration of adsorbed oxygen ions involved in the reaction and further reduced the effective electron mobility. The reduction in the electron mobility of the ITO resulted in a decrease in the gas-sensing response of the ITO gas sensor. The reaction process can be explained as Equations (3)–(5).
(3)$$O_{2}+2e^{-}\to 2O^{-}$$
(4)$$ITO\left(In_{2}O_{3}\right)+12SO_{2}\to In_{2}S_{3}+9SO_{3}$$
(5)$$ITO\left(SnO_{2}\right)+SO_{3}+O^{-}+e^{-}\to SnSO_{4}+2O^{-}$$

Based on the above analysis, the reaction between SO_2_ and the ITO film could cause damage to the materials, leading to abnormal changes in the trend of gas-sensitive reactions. Two sets of samples under the same process conditions (In_2_O_3_/SnO_2_ ratios of 85 wt%:15 wt%) and the same test conditions (working temperature of 200 °C) were used for verification. In Figure 7a, a low concentration of 5 ppm SO_2_ was first used, and its sensitivity was found to be 1.24. After the sensor was restored, the test baseline increased, indicating damage to the ITO material and an increase in the resistance. By reusing the high concentration of 50 ppm SO_2_, the sensitivity became 1.16, showing a decreasing trend. Once the sensor was restored again, the test baseline was further elevated, indicating that material damage was exacerbated. Finally, using the SO_2_ of 5 ppm for testing again, the sensitivity was only 1.06, which was significantly lower than the initial test result, strongly proving that the SO_2_ gas reaction did indeed cause damage to the material. The detailed gas-sensing characteristic parameters are presented in Table 4. Figure 7b shows the results of the other set of gas sensors, which were first reacted with a high concentration of 50 ppm SO_2_, then reacted with a low concentration of 5 ppm SO_2_ after recovery, and finally validated with 5 ppm again. All the gas-sensitive parameters are presented in Table 5, which fully demonstrates that ITO could seriously damage the gas-sensitive material when testing SO_2_ gas.

### 3.6. Gas Selectivity and Temperature Dependence for ITO-Based Sensors

Figure 8 shows the results of ITO-based gas sensors for the selectivity of NO_2_ and SO_2_ mixed gases. Due to the damage of SO_2_ gas to ITO materials, we chose a fixed SO_2_ concentration of 5 ppm and conducted gas-sensitivity testing by adjusting the NO_2_ gas concentration. Though the NO_2_ concentration (2.5 ppm) was lower than, equal to (5 ppm), or higher than (10 ppm) the SO_2_ concentration, all the response trends presented the NO_2_ response pattern, indicating that ITO materials have priority reactions with NO_2_ gas.

At the same time, the temperature dependence of gas reactions was also characterized, as shown in Figure 9. In the atmosphere of NO_2_ gas (5 ppm), the sensitivity of ITO-based gas sensors showed a parabolic trend with increasing temperature, indicating that there was an optimal working temperature. In the atmosphere of SO_2_ gas (5 ppm), due to the damage to the ITO material, the sensitivity of the sensor continuously increased with the increase in working temperature, showing a linear relationship.

### 3.7. Effect of Humidity on Sensor Performance

For acid rain gases, it is crucial to investigate the effect of humidity during gas sensor detection on device performance. The testing environment for this work was in a clean room with constant temperature and humidity (25 °C, 35%RH), so the starting point of humidity was 35%RH, which increased by 20% in sequence and reached 100%RH. Taking NO_2_ as the testing gas at a concentration of 100 ppm and a working temperature of 240 °C, it can be seen from the results in Figure 10 and Table 6 that the sensitivity of the gas sensor decreased from 2.42 to 1.49 with the humidity increase, indicating a decreasing trend. The response time and recovery time both continued to increase. On the one hand, an increase in humidity meant an increased number of water molecules in the relative space, which hindered the adsorption between gas molecules and sensitive materials. On the other hand, the solubility of NO_2_ in water to some extent reduced the concentration of tested NO_2_ gas. But overall, gas sensors based on ITO materials have obvious gas response characteristics and can work effectively in environments with humidity up to 100%RH.

## 4. Conclusions

In summary, this paper investigated the gas-sensing properties of ITO-based gas sensors with different indium/tin ratios for NO_2_ and SO_2_. The increase in the SnO_2_ ratio contributed to the increase in the specific surface area and the concentration of oxygen vacancies in the ITO film, which greatly improved its gas-sensing performance. By increasing the SnO_2_ ratio from 10% to 15%, the sensitivity was doubled for the detection of 5 ppm NO_2_, and both response and recovery time were improved. It was found that ITO-based gas sensors showed a decrease in sensitivity with increasing concentration for the detection of SO_2_. By analyzing the mechanism, the adsorption between the ITO film and SO_2_ molecules was chemical adsorption. Detailed discussions have been conducted on the gas selectivity, temperature dependence, and environmental humidity of ITO-based gas sensors. Gas-sensing devices prepared based on ITO materials can effectively monitor the concentration of acidic gases and prevent the formation of acid rain.

## Figures and Tables

**Figure 1 sensors-24-01286-f001:**
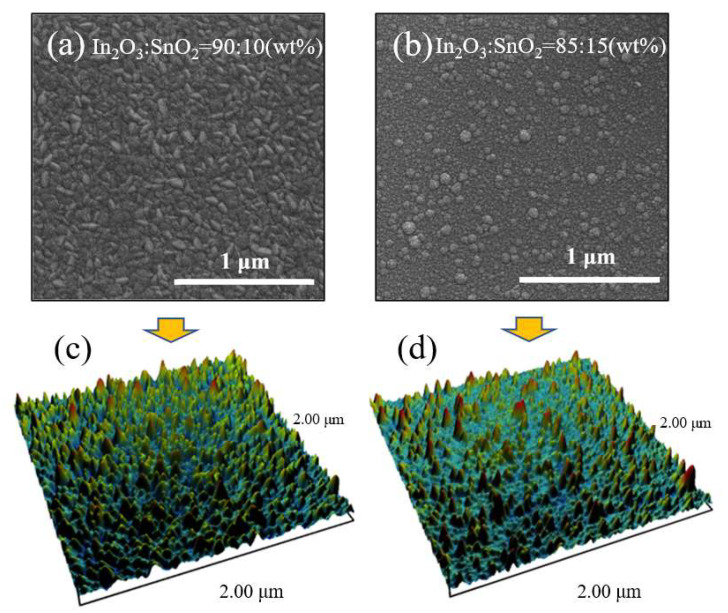
The SEM micrographs and AFM characterization results of ITO thin film with In_2_O_3_/SnO_2_ ratio of (**a**,**c**) 90:10 and (**b**,**d**) 85:15.

**Figure 2 sensors-24-01286-f002:**
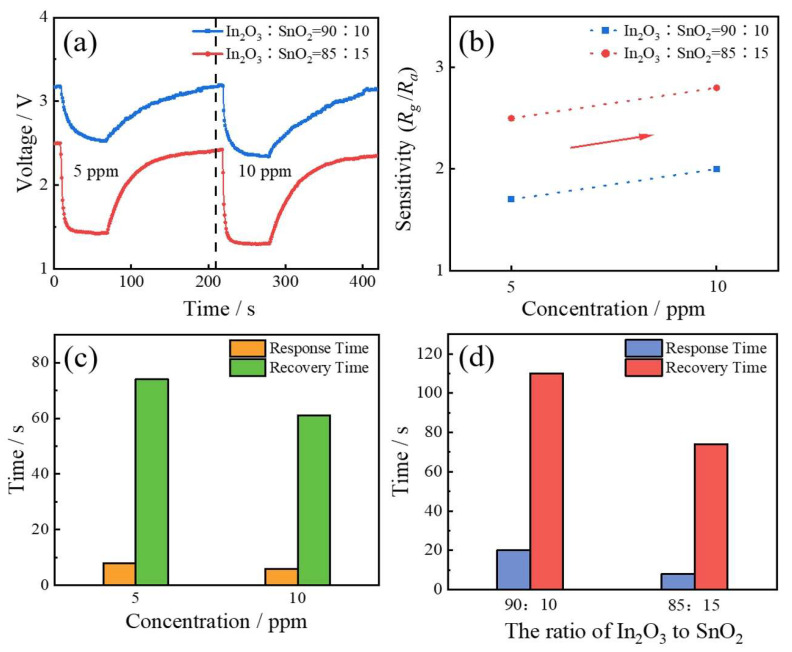
(**a**) Characterization curves of ITO gas sensors with different ITO ratios for detecting NO_2_. (**b**) Sensitivity of ITO gas sensors with different indium/tin ratios for the detection of 5 ppm and 10 ppm NO_2_. (**c**) Response and recovery time of ITO gas sensors with an indium/tin ratio of 85:15 for detecting NO_2_ at different concentrations. (**d**) Response and recovery time of ITO sensors with different In_2_O_3_/SnO_2_ ratios for detecting NO_2_ at the concentration of 5 ppm.

**Figure 3 sensors-24-01286-f003:**
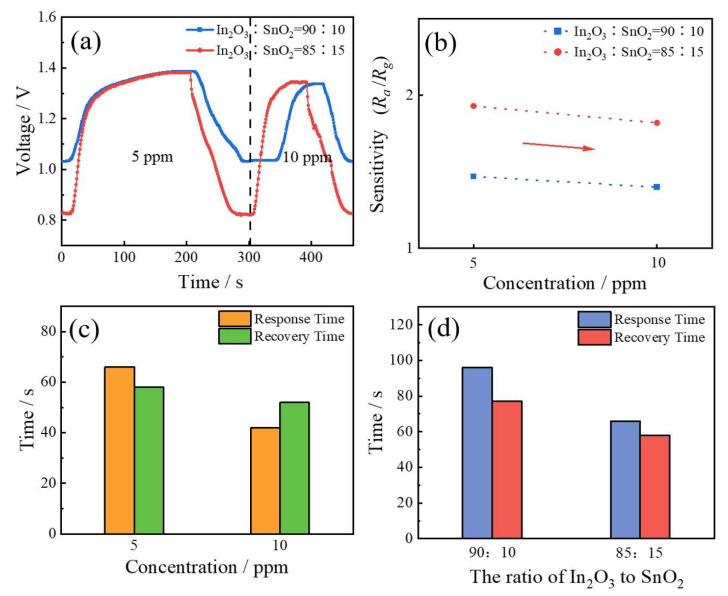
(**a**) Response curves of ITO-based gas sensors with different ITO ratios for the detection of SO_2_. (**b**) Sensitivity of ITO-based gas sensors with different indium/tin ratios for 5 ppm and 10 ppm SO_2_. (**c**) Response and recovery time of ITO-based gas sensors with an In_2_O_3_/SnO_2_ ratio of 85:15 for detecting SO_2_ at different concentrations. (**d**) Response and recovery time of ITO-based sensors with different indium/tin ratios for detecting 5 ppm SO_2_.

**Figure 4 sensors-24-01286-f004:**
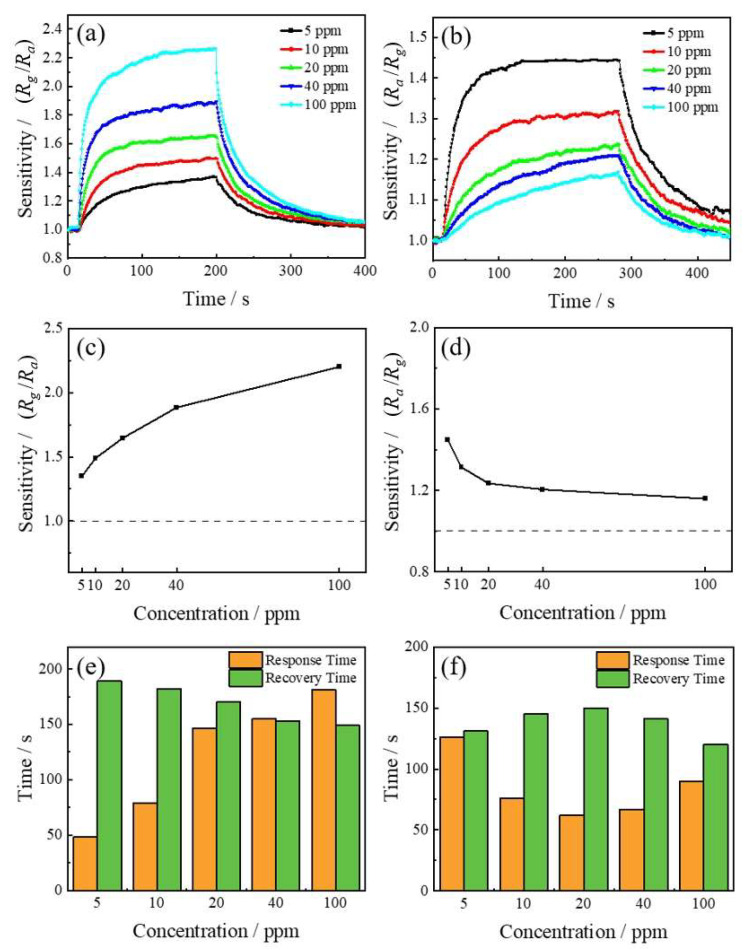
The response curve of ITO-based gas sensor in 5–100 ppm (**a**) NO_2_ and (**b**) SO_2_. The sensitivity of ITO-based gas sensors in detecting different concentrations of (**c**) NO_2_ and (**d**) SO_2_. The response time and recovery time of (**e**) NO_2_ and (**f**) SO_2_ with a concentration of 5–100 ppm.

**Figure 5 sensors-24-01286-f005:**
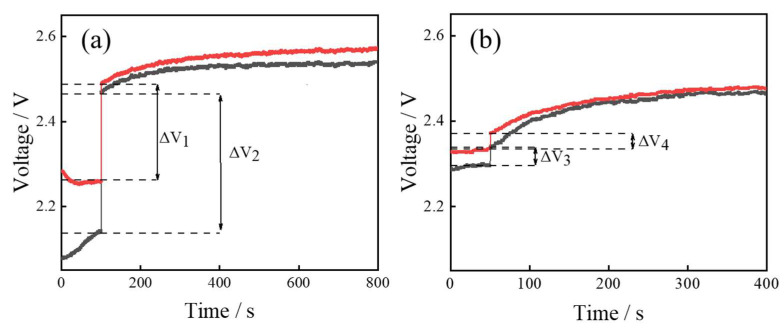
Static gas-sensitivity test curve of ITO gas sensor in (**a**) 5 ppm and (**b**) 10 ppm SO_2_ environment.

**Figure 6 sensors-24-01286-f006:**
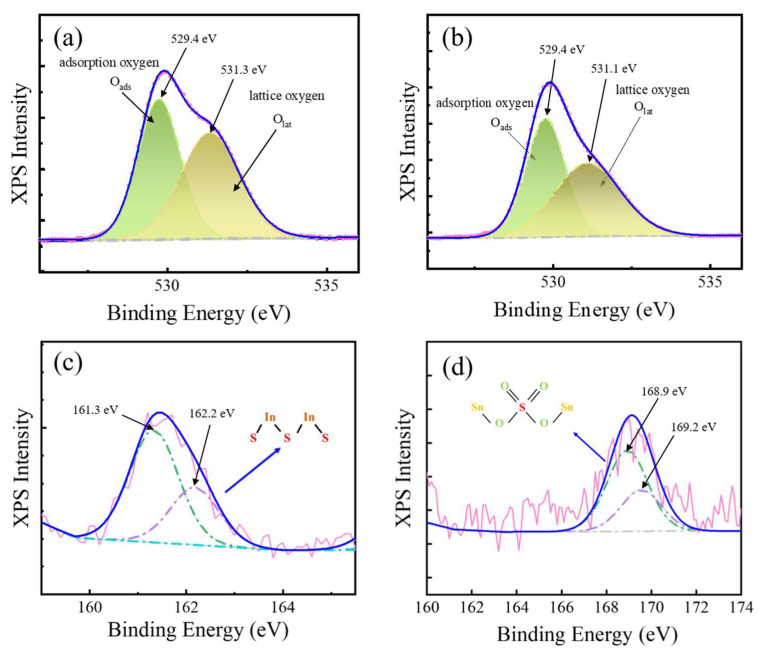
Single-peak fits of O1s in ITO with different In_2_O_3_/SnO_2_ ratios: (**a**) 90:10, (**b**) 85:15. XPS spectra of S2p in ITO after participating in the reaction with (**c**) 5 ppm SO_2_ and (**d**) 10 ppm SO_2_.

**Figure 7 sensors-24-01286-f007:**
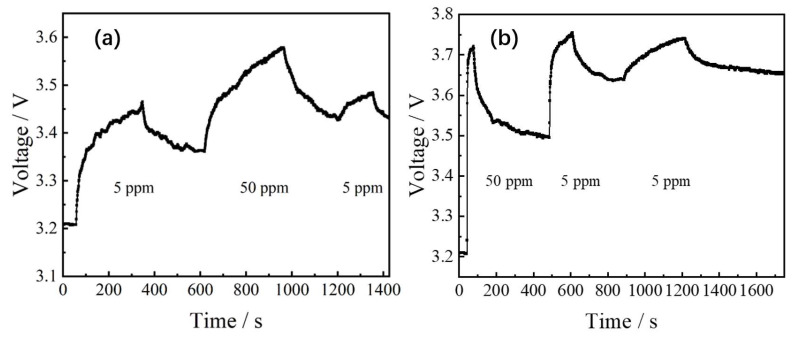
The reaction concentration of SO_2_ gas is (**a**) initially low and then high, (**b**) initially high and then low.

**Figure 8 sensors-24-01286-f008:**
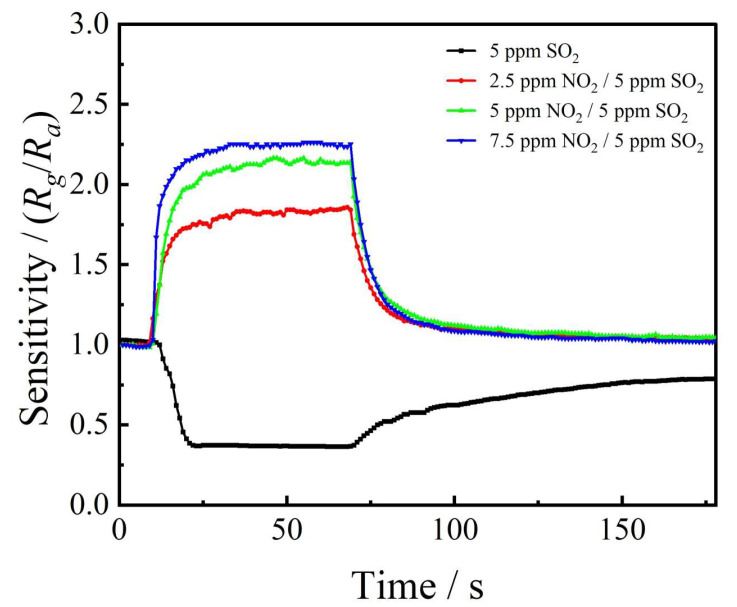
The gas sensitivity of NO_2_ and SO_2_ mixed gases.

**Figure 9 sensors-24-01286-f009:**
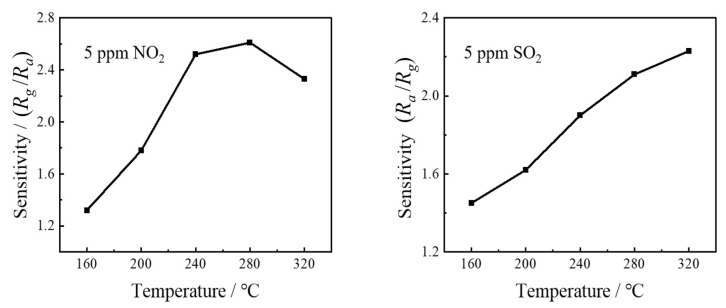
The effect of temperature on gas-sensing characteristics.

**Figure 10 sensors-24-01286-f010:**
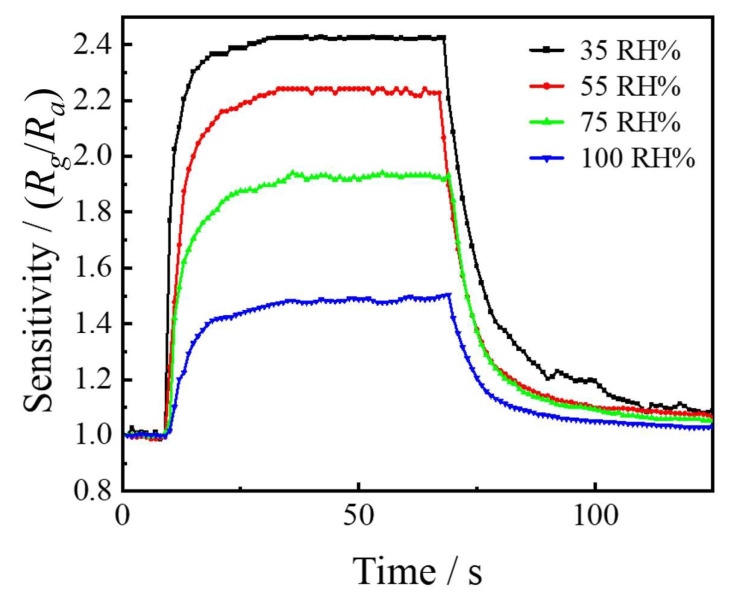
The sensitivity of ITO-based gas sensors under different humidity.

**Table 1 sensors-24-01286-t001:** The comparison of gas-sensitivity characteristics for NO_2_.

Sensing Material	Working Temperature/°C	Concentration/ppm	Sensitivity/(*R_g_*/*R_a_*)	Response/Recovery Time/s	Ref.
ZnO	200	6	18.4	300/200	[23]
rGO/SnO_2_	125	3	53.6	16/63	[24]
ZnO	230	4	6.2	18/65	[25]
ZnSe/SnO_2_	160	2.4	6.94	64/52	[26]
Ni/WO_3_/rGO	240	10	2.95	~420/-	[27]
ITO(In_2_O_3_-SnO_2_)	100	50	100	30/40	[28]
ITO(In_2_O_3_-SnO_2_)	150	10	~5	~480/~7920	[29]
ZnO-ITO(In_2_O_3_-SnO_2_)	160	70	~270	~82/~26	[7]
**ITO(In_2_O_3_-SnO_2_)**	**240**	**5**	**2.5**	**8/74**	**This work**

**Table 2 sensors-24-01286-t002:** The comparison of gas-sensitivity characteristics for SO_2_.

Sensing Material	Working Temperature/°C	Concentration/ppm	Sensitivity/(*R_a_*/*R_g_*)	Response/Recovery Time/s	Ref.
NiO/SnO_2_	180	500	~56	80/70	[30]
V_2_O_5_/SnO_2_	240	500	1.5	106/115	[31]
MgO/SnO_2_	280	500	3.17	59/52	[32]
Cu/SnO_2_	250	6	91.51	270/901	[33]
Au/La_2_O_3_/ZnO	260	1	1.41	~90/~60	[34]
Au/SnO_2_	200	5	4.9	~100/~70	[35]
NiO/SnO_2_	240	10	1.7	~30/~40	[36]
Ru/Al_2_O_3_/ZnO	350	5	1.12	~60/~360	[37]
**ITO(In_2_O_3_-SnO_2_)**	**240**	**5**	**1.8**	**66/58**	**This work**

**Table 3 sensors-24-01286-t003:** The amount of oxygen in different ITO films.

Indium/Tin Ratio	Adsorbed Oxygen (O_ads_)/C	Lattice Oxygen (O_lat_)/C	Ratio of O_ads_/O_lat_
90%:10%	224,753.7	239,619.1	0.938
85%:15%	256,602.3	269,028.7	0.954

**Table 4 sensors-24-01286-t004:** The gas-sensitivity characteristics from Figure 7a.

Test No.	Concentration/ppm	Sensitivity/(*R_a_*/*R_g_*)	Response/Recovery Time/s
1	5	1.24	198/154
2	50	1.16	241/177
3	5	1.06	120/62

**Table 5 sensors-24-01286-t005:** The gas-sensitivity characteristics from Figure 7b.

Test No.	Concentration/ppm	Sensitivity/(*R_a_*/*R_g_*)	Response/Recovery Time/s
1	50	1.62	8/190
2	5	1.30	81/147
3	5	1.11	244/307

**Table 6 sensors-24-01286-t006:** Sensor characteristics with different humidity.

Relative Humidity/ (% RH)	Sensitivity/(*R_g_*/*R_a_*)	Response/Recovery Time/(s)
35	2.42	5/34
55	2.24	10/27
75	1.92	12/34
100	1.49	15/36

## Data Availability

The data presented in this study are available on request from the corresponding author.

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
