# Peer review of "Acidic Gas Determination Using Indium Tin Oxide-Based Gas Sensors"

_sensors, 2024, doi:10.3390/s24041286_

Round 1

Reviewer 1 Report

Comments and Suggestions for Authors

I would reccomend the major revision of the manuscript before the acceptance for the publication, according to the following points:

1. Please go through the comments in the attached file in order to clarify some ambigous issusein the manuscript text.

2. The comparison of the performance of sensing material designed by Authors with other NO2 and SO2 sensors is missing.

3. The information of the operating temperature of the proposed sensors is missing.

4. What about the selectivity and long-term stability of the sensing materials? Did Authors perform some tests in this matter?

Reviewer 2 Report

Comments and Suggestions for Authors

Dear authors,

Your manuscript presents an advancement in gas sensing technology, specifically focusing on indium-zinc oxide (ITO) sensors for detecting SO2 and NO2 gases. A significant enhancement in the sensitivity towards NO2 was observed when the SnO2 ratio in ITO was increased to 5%. This modification also led to faster response and recovery times in NO2 detection. However, the study found a decrease in sensitivity for SO2 at higher concentrations. The research delved into the mechanisms of gas sensing, revealing that NO2 adsorbs through physisorption while SO2 undergoes a chemical reaction with the surface of ITO. These findings underscore the potential of ITO sensors in environmental monitoring, particularly in tracking acid rain, and pave the way for further development of high-efficiency gas sensors. However, I regretfully have to reject this manuscript for the following reasons:

1. The number of survey samples is too small (2 samples) with In2O3/SnO2 ratios of 90/10 and 85/15.

2. There is absolutely no data to prove the formation of the ITO phase according to the manufactured ratios.

3. The gas sensitivity data in section 3.2 only has 2 data points. There are no data on membrane resistance.

4. There is no survey data on the effect of temperature on sensor performance.

5. There are no humidity figures included in these measurements. There needs to be an investigation into the influence of humidity on the sensor because the sensor's target is acid rain.

In short, this is an incomplete study and needs additional data before submission.

Round 2

Reviewer 1 Report

Comments and Suggestions for Authors

Dear Authors,

thank you for improving your manuscript according to the Reviewers'comments. I am happy to accept your work for publication.

Author Response

Thanks for your recognition. Your encouragement is the driving force for us to keep moving forward.

Reviewer 2 Report

Comments and Suggestions for Authors

Despite my earlier refusal to publish the manuscript, I am grateful to the authors for their efforts in seeing it through to completion. Numerous distinct semiconductor oxides have been used in studies on the sensitivity properties of gases such as SO2 or NO2. Numerous investigations have demonstrated that these gases are detectable at room temperature. Have the authors examined the gas-sensing characteristics of the fabricated ITO material at this temperature, given the low ITO film resistance? The NO2 and SO2 gas sensitivity characteristics of the manufactured ITO membrane are not compared to those of other ITO membranes in Tables 1 and 2, according to the author. Why did the author of these tables decide that 240oC was the ideal operating temperature for the sensor? Acid rain is the sensor's research subject. Why did the author investigate the gas sensitivity characteristics of SO2 and NO2 at 35% humidity? The gas-sensing characteristics of ITO membranes at higher humidity levels—even near 100%—need to be further explored by the author.
